# 1 Winter sea ice edge shaped by Antarctic Circumpolar Current

# 2 pathways.

- Hugues Goosse<sup>1</sup>, Stephy Libera<sup>1</sup>, Alberto C. Naveira Garabato<sup>2</sup>, Benjamin Richaud<sup>1</sup>,
- Alessandro Silvano<sup>2</sup>, Martin Vancoppenolle<sup>3</sup>
- <sup>1</sup>Earth and Life Institute, Université catholique de Louvain, Louvain-la-Neuve, Belgium
- <sup>2</sup>Ocean and Earth Science, National Oceanography Centre, University of Southampton, Southampton, UK.
- <sup>3</sup>Sorbonne Université, Laboratoire d'Océanographie et du Climat (LOCEAN-IPSL), CNRS/IRD/MNHN, Paris,
- France
- *Correspondence to*: Hugues Goosse (hugues.goosse@uclouvain.be)

### 10 Abstract

The Antarctic Circumpolar Current (ACC) is often considered a natural barrier for the northward 12 expansion of the Antarctic sea ice, but the underlying processes remain little explored. Here, we focus 13 on the main fronts of the ACC — as a measure of the current system's path — to study how they 14 influence the mean state of sea ice. We find that the latitude of all ACC fronts as a function of longitude 15 shows a correlation above 0.85 with the climatological mean latitude of the winter sea ice edge, 16 indicating a strong link across all sectors of the Southern Ocean. Among the ACC fronts, the Polar Front 17 is identified as the best indicator for studying the ACC's influence on sea ice, as it marks a distinct 18 transition in upper-ocean water mass properties and is consistently found north of the sea ice edge. 19 The distance between the Polar Front and the sea ice edge decreases when the Polar Front lies farther 20 south, due to the presence of warmer waters at higher latitudes. These warmer waters enable efficient 21 heat transport toward the ice edge and constitute a barrier to sea ice expansion, via two main 22 mechanisms. First, the ocean carries heat from the Polar Front toward the pole, in particular through 23 the contribution of mesoscale ocean eddies formed downstream of major underwater topographic 24 features. Second, warmer oceanic surface waters near the front heat the atmosphere above, which 25 then carries this heat poleward towards the ice, especially in regions with more southward-directed 26 winds. Since the Polar Front's path is largely shaped by topographic barriers, these results indicate why 27 the position of the winter sea ice edge is strongly constrained, under current conditions, by 28 bathymetry.

## 1 Introduction

- In contrast to the Arctic, where the sea ice cover in winter is strongly constrained by the coastal
- geometry and can only expand southward in a few sectors (Bitz et al., 2005; Eisenman, 2010), the
- winter Antarctic sea ice extent is not limited to the north by any continent. The Antarctic Circumpolar
- Current (ACC) is often considered a natural boundary for the expansion of the winter sea ice in the
- Southern Ocean, but the nature of the link remains largely unexplored and the underlying mechanisms
- are only partially understood (Martinson, 2012; Nghiem et al., 2016).
- Ideally, in the absence of continents, the freeze and melt of Antarctic sea ice would be determined by
- the seasonal warming and cooling of the surface ocean driven by changes in insolation; in this case,
- the seasonal cycle of sea ice cover would be governed solely by latitude (Roach et al., 2022). Indeed,
- Antarctic sea ice cover expansion is primarily controlled by thermodynamic processes that cool the
- Antarctic Surface Water (AASW) in fall and winter, removing the heat stored in summer until the mixed
- layer temperature reaches the freezing point and sea ice forms (Bitz et al., 2005; Eayrs et al., 2019;
- Roach et al., 2022; Himmich et al., 2023). Sea ice is then transported equatorward, towards

- increasingly warmer waters that tend to melt the ice. This sea ice transport plays a secondary role in
- the inner pack, but makes a larger contribution at the ice edge (e.g., Bitz et al., 2005; Holland and
- Kimura, 2016; Eayrs et al., 2019; Himmich et al., 2023).
- However, the observed location of the winter sea ice edge does not follow a uniform circumpolar
- latitude in the Southern Ocean. These deviations in ice edge latitude may be explained by
- convergences in atmospheric and oceanic heat flux transport, which modulate heat storage and loss
- in the ocean mixed layer (Bitz et al., 2005). In this framework, the ACC may influence sea ice advance
- and the winter ice edge position by modulating heat transport towards high latitudes.
- The ACC is a strong eastward current that connects the three main ocean basins (see for instance
- Rintoul et al., 2001; Rintoul, 2018 for reviews). The absence of land or topographic barriers in the
- latitudinal band of the Drake Passage implies that no net meridional geostrophically balanced flow can
- occur in the upper ocean between roughly 56-58° S, limiting exchanges between subtropical and polar
- oceans. As a result of geostrophic balance, the ACC's eastward flow is associated with a steep rise of
- isopycnals (surfaces of constant density) toward the south. The sloping isopycnals and strong currents
- induce intense mesoscale eddy activity in the ACC (Frenger et al., 2015). These eddies are an essential
- element of the dynamical balance of the current, vertically transporting the momentum to the bottom
- of the ocean, and are responsible for a large part of the exchanges across the ACC, and specifically of
- the meridional heat flux (Rintoul et al., 2001; Naveira Garabato et al., 2011; Naveira Garabato et al.,
- 2016; Morrison et al., 2016, Rintoul, 2018; Liu et al., 2024).
- The path of the ACC is strongly controlled by the bathymetry and is characterized by strong fronts,
- separating water masses with different characteristics (Orsi et al., 1995; Rintoul, 2001; Langlais et al.,
- 2011; Graham et al., 2012; Rintoul, 2018; Nghiem et al., 2016, de Boer et al., 2022; Wilson et al., 2022).
- The traditional view is to consider three main circumpolar fronts within the ACC, which split into
- different branches or merge at some locations (Orsi et al., 2015; Sokolov and Rintoul, 2009; Langlais et
- al., 2011; Kim and Orsi, 2014; Pauthenet et al., 2017; Park et al. 2019). From south to north, we have
- the Southern ACC front (sACCF), the Polar Front (PF) and the Subantarctic Front (SAF) (Fig. 1).
- Additionally, the Southern Boundary (SBdy) of the ACC delimits the eastward circumpolar flow and,
- north of the ACC, the Subtropical Front (STF) marks the separation with subtropical waters.
- ACC fronts can be defined using criteria based on temperature and salinity gradients, usually from
- subsurface properties as variability and noise are weaker than at surface (Orsi et al., 1995). However,
- the fronts mark transitions through the entire water column. Due to the thermal wind relation, the
- boundaries between water masses are associated with intense currents or jets, which are responsible
- for the majority of the ACC transport (Orsi et al., 1995; Rintoul, 2001; Sokolov and Rintoul, 2009). The
- correspondence between the sea surface height (SSH) gradients linked to those jets and specific SSH
- values have led to another method for identifying frontal positions using a judicious choice of
- circumpolar SSH contours (Sokolov and Rintoul, 2009; Sallée et al., 2008; Kim and Orsi, 2014; Park et
- al., 2019) (See section 2a for more details on frontal definitions). However, this identification method
- suffers from the fact that jets and fronts are not necessarily circumpolar. Instead, they evolve rapidly,
- and jets are not always clearly associated with temperature and salinity fronts, as indicated by satellite-
- based and modelling studies (Thomson and Sallée, 2012; Chapman et al., 2017; Chapman et al., 2020).
- Because of the ACC's complex dynamics, it is not obvious to identify which ACC elements, and which
- mechanisms, could be linked with the seasonal expansion of Antarctic sea ice. First, oceanic
- temperatures well above freezing can promote melting or hinder freezing. Therefore, the Polar Front
- could provide a physically sensible limit to sea ice expansion, as it marks the boundary between cold
- and fresh Antarctic Surface Waters and warmer waters to the north (Orsi et al., 1995; Kim and Orsi,

2014). The PF is also better defined and stronger than the sACCF and SBdy at most longitudes, being associated in general with more intense currents. Sea ice pushed by strong winds crossing the PF would melt rapidly upon contact with warm waters, and would not persist for long. However, the PF lies several degrees north of the winter ice edge in nearly all sectors of the Southern Ocean, so if it does constrain sea ice expansion, it must do so through a subtler mechanism. The SAF is located even farther north of the sea ice edge. It is therefore not expected to directly impact sea ice and is not addressed here.

Second, the SST patterns associated with the ACC fronts affect the ocean-atmosphere heat flux north of the ice edge. This could modify the magnitude of the southward atmospheric heat transport, indirectly affecting the melting or freezing of sea ice (Blanchard-Wrigglesworth et al., 2021; Kusahara and Tatebe, 2025). Third, the fronts are also characterized by subsurface temperature gradients. In particular, the sAACF and the SBdy have a weak a surface expression, with cold Antarctic Surface Waters being present everywhere south of the PF (Orsi et al., 1995; Kim and Orsi, 2014; Pauthenet et al., 2018), but they could influence the ice pack by modulating the vertical heat exchanges with warmer subsurface waters. However, these vertical exchanges are controlled by many processes and feedbacks (Martinson et al., 1990). This makes the identification of a direct link between the position of the sACCF or SBdy and the ice edge challenging, as illustrated by the presence of those fronts north or south of the ice edge in different regions. Fourth, the strong zonal currents in the ACC could push sea ice eastward, limiting its northward movement, and the oceanic currents could affect the sea ice cover through their role on meridional heat transport. Finally, eddy formation within the ACC contributes to horizontal and vertical heat exchanges (Dufour et al., 2015; Hausmann et al., 2017, Gupta et al., 2020; Manucharyan and Thompson, 2022; Ferola et al., 2023). We might expect that eddies generated in warm waters north of the PF and translated southward would limit ice formation or would increase melt, at least locally.

This brief overview indicates that considering the ACC as a boundary for sea ice expansion seems natural at first sight, but it is less straightforward to identify the mechanisms by which the ACC would exert this influence and how such influence might vary between different regions. To our knowledge, no systematic, circumpolar analysis of the link between sea ice expansion and the ACC currents or fronts has been conducted to date. One noticeable exception is for the region of the Pacific-Antarctic Ridge between 160°W–135°W, where topography strongly controls the position of the all the fronts and thereby sets the location of the ice edge (Roach and Speer, 2019; Ferola et al., 2023).

Studying the interactions between the ACC and sea ice is a very broad task. We will focus here only on the mean state of the system, not on interannual variability. This has the advantage that we can use relatively robust and easy-to-interpret definitions of the fronts, based on long-term collections of oceanographic observations, as well as the SSH contour method, which is less robust for interannual variability (Chapman et al., 2020). Further, the interannual variability of sea ice is mainly driven by atmospheric processes (Bitz et al., 2005). Removing this atmospheric contribution to identify the potential influence of variations within the ACC is a specific objective that we defer to future work. We will also consider only the winter ice edge (defined here by its September location), as the summer sea ice limit lies far too south of the ACC, and is thus less directly affected by ACC characteristics.

We put forward the hypothesis that the main influence of the ACC on the mean position of the winter sea ice edge is robust enough to be identified, despite uncertainties in the available datasets, using a relatively simple methodology. Another objective of the simple diagnostics presented here is to highlight the dominant elements that could be blurred by the additional complexity introduced by discussing in greater detail all the processes at play. The methodology and selected observations are

described in section 2. The main results are discussed in section 3, before conclusions on the role of the ACC in shaping Antarctic sea ice expansion are offered in section 4.

### 2 Methodology

a/ Definition of ACC fronts.

We use here definitions of ACC fronts that provide robust, circumpolar positions, fixed in time, that we will relate (in section 3) to the mean location of the winter sea ice edge. A first option is to use criteria based on specific values of oceanic temperature and salinity obtained from in situ profiles. Exact values vary slightly between authors or regions, introducing some differences between studies, but the general characteristics remain similar. Here, we have taken the classical indicators of Orsi et al. (1995) and their estimates on front position based on in situ observations available up to 1990 (Table 1). Although additional elements must be specified for a complete definition (see Table 3 of Orsi et al. 1995), the SAF is mainly characterized by the downwelling of low-salinity waters sourced from the south, and 4-5°C isotherm located at a depth around 400 m. The PF is located at the northernmost position of the 2°C temperature minimum of Winter Water at depths shallower than 200 m. The PF also marks the transition from a surface stratification based on salinity in the south, to stratification based on temperature in the north (Orsi et al., 1995; Kim and Orsi, 2014). The sACCF is defined by the southern limit of the 2°C isotherm at the subsurface temperature maximum corresponding to the presence of Circumpolar Deep Water. Finally, the SBdy can be taken as the location where Upper Circumpolar Deep Water enters the mixed layer (Kim and Orsi, 2014).

Table 1: Some key characteristics used to define the ACC fronts, simplified from Orsi et al. (1995)

| Polar Front (PF)           | Northernmost position of the 2°C temperature minimum of Winter   |
|----------------------------|------------------------------------------------------------------|
|                            | Water at depths shallower than 200 m                             |
| Southern ACC front (sACCF) | Southern limit of the 2°C isotherm at the subsurface temperature |
|                            | maximum                                                          |
| Southern Boundary of the   | Location where Upper Circumpolar Deep Water enters the mixed     |
| ACC (SBdy)                 | layer                                                            |

A second option is to select values of dynamic topography that correspond to the frontal positions at locations where they are well defined from oceanographic data (Kim and Orsi, 2014; Park et al., 2019). This is based on the assumption that fronts are associated with strong currents, as discussed in the Introduction. Maps of dynamic topography can be derived from the sea surface height observed from satellites, and the selected contours provide circumpolar positions of the fronts. We will use here the estimate of Park et al. (2019) based on satellite observations over the 1993-2012 period.

A broad correspondence between the two definitions is expected on physical grounds, and due to the selection of the dynamical contours that match some in situ estimates of the frontal locations. These two definitions are thus linked but they are based on independent data sets (i.e. in situ measurements of ocean properties for Orsi et al., 1995, and satellite observations of SSH for Park et al., 2019). Performing the analysis for both frontal indicators thus provides a test of the robustness of our results, and of the sensitivity of our conclusions to the way the fronts are defined.

Other methods have been proposed to detect fronts based on local SSH and SST gradients, or from the automatic, objective identification of strong differences between nearby oceanic profiles (Thompson and Sallée, 2012; Graham et al., 2012; Chapman, 2017; Pauthenet et al., 2018; Thomas et al., 2021; Denes et al., 2025). Such local definitions of fronts have the advantage of not requiring an a priori choice of the number of fronts or of a fixed circumpolar threshold to identify them. The fronts can also

- be discontinuous and are thus not identified in regions where there is no sharp boundary between
- water masses or where no clear jet exists, as in global methods imposing circumpolar continuity. They
- are also more adapted to study the time variability of front position. However, those local definitions
- lead to features that are in general more difficult to interpret and this is the reason they have not been
- chosen here (see Chapman et al., 2020 for a comparison of the advantages and disadvantages of local
- and global definitions of fronts).
- b/ Observations and model results
- The sea ice concentration is obtained from the EUMETSAT OSI SAF (OSI-450 and OSI-430-b; OSI SAF
- 2017; Lavergne et al., 2019) and the near surface air temperature and 10-m wind speed from the ERA5
- reanalysis (Hersbach et al., 2020a). The location of the winter ice edge is defined as the 15% limit of
- sea ice concentration in September averaged over the period 1979-2023, following the algorithm of
- Eisenman (2010). Climatological oceanic temperature and salinity come from the estimates of
- Yamazaki et al. (2025). The bathymetry is based on ETOPO1 (Amante and Eakins, 2009). The eddy
- kinetic energy is taken from Auger et al. (2022), who derived sea level anomaly over the Southern
- Ocean, including sea ice covered regions, from 2013 to 2019 using satellite observations from multiple
- sources.
- As no estimate based on observations is available, we use the oceanic sensible heat flux at the base of
- the sea ice over the period 1979-2023 computed in the simulation of Richaud et al. (2025), performed
- with the Ocean model NEMO4.2 at a resolution of ¼° and driven by ERA5 reanalysis. The NEMO model
- (Madec, G. and the NEMO System Team, 2023) includes the OPA ocean model (Océan PArallélisé)
- coupled to the SI<sup>3</sup> sea ice model (Vancoppenolle et al., 2023). For consistency, we also analyse the
- oceanic heat flux at the sea surface from the same simulation.
- The datasets correspond to different periods, but we assume that each of them is long enough to
- represent the mean conditions in the Southern Ocean over the recent decades and that the differences
- induced by the interannual variability and multi-annual trends are smaller than the signal we wish to
- extract.

### 3. Results

- a/ Links between the position of the ACC fronts and the location of the sea ice edge in winter
- Figure 1 shows the position of the ACC fronts, providing a simple and direct representation of the path
- and extent of the ACC that is compared to the location of the winter ice edge. The bathymetry is also
- indicated, as it controls the position of the currents in the ACC and of the fronts, guiding them or
- deflecting their path, in particular close to the South Scotia Arc (300°-320°E), the Kerguelen plateau
- (70-80°E), the Southeast Indian ridge (95°E-140°E) and the Antarctic-Pacific ridge (160°-210°E) (Orsi et
- al. 1995; Rintoul, 2001; Langlais et al., 2011; Rintoul, 2018; Nghiem et al., 2016; Wilson et al., 2022).
- After those ridges, the path of the ACC is generally deflected northward, while in more flat areas, it
- tends to go southward (Rintoul, 2001; Patmore et al., 2019; Jouanno and Capet, 2020). Additionally, in
- deeper regions, the fronts are generally clearly distinct but, when the flow encounters a large
- topographic barrier, the fronts can merge, for instance around 200°E (Sallée et al., 2011; Thompson
- and Sallée, 2012; Chapman, 2017; Rintoul, 2018).
- The position of the fronts is similar between the hydrography-based definition of Orsi et al. (1995) and
- the dynamical height-based definition of Park et al. (2019), especially for the PF. Slightly larger
- differences are present for the sACCF and SBdy, such as around 60°E, where they are both very close
- to the continental slope according to Orsi et al. (1995) but further offshore in Park et al. (2019). The PF

is always located to the north of the winter ice edge, while the sACCF and the SBdy can be located north or south of the winter ice edge. This suggests that the influence of the PF on the northward expansion of sea ice is more consistent between regions than for the sACCF and the SBdy. The sACCF and the SBdy are close to the winter ice edge in several regions, in particular over the Antarctic-Pacific ridge between 160°E and 210°E and near to the Scotia Arc between 300°E and 320°E, but further away in some other sectors. The distance between the location of these two fronts and the ice edge is generally larger when using the front definitions from Orsi et al. (2015).

While it is clear that ocean currents and fronts can impact sea ice, sea ice processes in return modify oceanic temperature and salinity, which impact the large-scale ocean state and thereby potentially the ACC fronts. Sea ice melting and freezing induces strong heat, salt, and freshwater fluxes (Martinson, 1990; Wilson et al. 2019), controlling vertical mixing as well as meridional transport of freshwater with consequence to water mass transformation (Abernathey et al. 2016; Pellichero et al. 2018). The sea ice drift also modulates the surface stress at the ocean surface compared to ice-free conditions. Consequently, sea ice processes affect the structure of the water column, specifically the stratification and the properties of the Winter Water, which is part of oceanic mixed layer in winter and below it in summer (Toole, 1981; Spira et al., 2024). The properties of the Winter Water and more generally of the water located just below the mixed layer (whose properties are influenced by the interactions with the mixed layer) are used to define the fronts (see Table 1). We cannot thus exclude the hypothesis that the agreement between the winter ice edge and frontal locations is driven in some regions by this impact of the sea ice on the ocean temperature and salinity rather than the impact of ocean on sea ice. This influence of sea ice is potentially larger for the sACCF and the SBdy, which are usually close to the ice edge, than for the PF, which is always located further north and thus not in contact with the sea ice. The sea ice processes also influence the oceanic properties at the latitude of the PF, in particular because of oceanic transport and advection of properties acquired in ice covered regions (Komuro and Hasumi, 2003; Haumann et al., 2016; Rintoul, 2021; Klocker et al., 2023a). However, the link with the position of the winter ice edge is not direct and the strong control of the location of the PF by the bathymetry mentioned above suggests that the contribution of sea ice is at most secondary. Finally, at a larger scale, the ice-ocean exchanges drive the oceanic density variations at high latitudes and thus the north-south density contrast that also contributes to the ACC transport (Rintoul et al., 2001; Klocker et al., 2023b) and thus potentially to the position and strength of the fronts.

The sACCF and SBdy have mainly subsurface expressions, as the cold Antarctic Surface Waters are present everywhere south of the PF. This is well illustrated by comparing sections in areas where the winter ice edge is to the north of the sACCF and SBdy and sections where the winter ice edge is south of these fronts (Fig. 2). In all cases, a cold surface layer with a temperature close to the freezing point is present in the top 100 m at high latitudes (corresponding to the area covered by sea ice in winter), with a sharp warming northward. The main differences occur at depth. In the first case, with the winter ice edge north of the two fronts, the temperature at depth is warmer, with temperatures higher than 2°C that can even reach the continental shelf. In contrast, the temperature at depth is colder in the second case, where the winter ice edge is southward of both fronts, with specifically the 2°C isotherm at depth northward of the ice edge. This is consistent with the frontal definition based on the criteria of Orsi et al. (1995) (Table 1), and indicates that the presence of relatively warm waters at depth does not impede the formation of sea ice at the surface if the stratification is strong enough to limit vertical exchanges. Besides, the presence of warm water at depth and its southward transport can have a dominant impact on other elements of the Southern Ocean system. For instance, a recent southward shift of the Circumpolar Deep Water, the sACCF and the SBdy has been observed off East Antarctica, potentially leading to additional melting of the ice shelves in this region (Yamazaki et al., 2021; Herraiz-Borreguero and Naveira Garabato, 2022).

While Figure 1 provides a straightforward visual comparison, displaying the latitude of the fronts and of the winter ice edge as a function of longitude allows for more quantitative analyses (Fig. 3). Off the Drake Passage (300°E), the ACC spreads equatorward in the Atlantic sector shifting all the fronts and winter ice edge location to lower latitudes. The ACC and the winter ice edge then converges poleward in the Indian and East Pacific sectors until 240°E where they move back northward. Next to this general common set-up, several local maxima in the latitude of the winter ice edge correspond to peaks in frontal positions (i.e. peaks in equatorward translation), such as around 80°E, 150°E and 200°E. This leads to a correlation with the winter ice edge latitude that is higher than 0.85 for all the fronts regardless of the frontal definition used (Table 2). The correlation remains high (with values larger than half of the peak correlation) for a spatial lag between the positions of the ice edge and of the front exceeding 30° of longitude, indicating that the observed high correlations have a large horizontal scale. The peak correlation is generally very close to a spatial lag of zero degrees of longitude.

This influence of the path of the ACC on the winter ice edge is clear in regions where the control of the ACC by topography is strong, but also between those topographic features as indicated by the good agreement at nearly all longitudes. The role of the position of the ACC is particularly well illustrated by comparing the Bellingshausen and Weddell Seas, respectively east and west of the Antarctic Peninsula. West of the Peninsula, the fronts are located relatively southward, with the ice edge close to the continent, while on the east, the deflection of the fronts imposed by the Scotia Arc allows a more northward position of the winter ice edge, which is very far from the continent.

Table 2. Correlations between the latitude of the winter sea ice edge as a function of longitude and the latitude of the fronts at the same longitude, between the distance (in degree of latitude) from the fronts to the winter sea ice edge and the latitude of the corresponding front, between the distance (in degree of latitude) from the fronts to the winter sea ice edge and the latitude of the winter sea ice edge and finally between the distance (in degree of latitude) from the fronts to the winter sea ice edge and the maginitude of the meridional 10-m winds at 60°S at the same longitude. The correlation are given for the PF, the sACCF and the SBdy, first following the defintion of Park et al. (2019) and second the definition of Orsi et al. (1995).

|                                               | PF   |      | sACCF |      | SBdy |       |
|-----------------------------------------------|------|------|-------|------|------|-------|
|                                               | Park | Orsi | Park  | Orsi | Park | Orsi  |
| Correlation of the latitude of the front with | 0.92 | 0.90 | 0.95  | 0.90 | 0.91 | 0.85  |
| the latitude of the ice edge                  |      |      |       |      |      |       |
| Correlation of the distance between the       | 0.82 | 0.83 | 0.88  | 0.82 | 0.86 | 0.72  |
| front and the ice edge with the latitude of   |      |      |       |      |      |       |
| the front                                     |      |      |       |      |      |       |
| Correlation of the distance between the       | 0.53 | 0.51 | 0.68  | 0.49 | 0.59 | 0.24  |
| front and the ice edge with the latitude ice  |      |      |       |      |      |       |
| edge                                          |      |      |       |      |      |       |
| Correlation of the distance between the       | 0.55 | 0.61 | 0.37  | 0.22 | 0.16 | -0.21 |
| front and the ice edge with the meridional    |      |      |       |      |      |       |
| winds at 60°S                                 |      |      |       |      |      |       |

b/ Analysis of the distance between the ACC fronts and the winter sea ice edge

After identifying this high correlation between the latitude of the fronts and of the winter ice edge, it is instructive to discuss the distance between the fronts and the ice edge. It is measured here by the difference in their position in latitude as a function of longitude. This distance varies substantially (Fig. 4). For instance, for the PF, it ranges from more than 10° of latitude in some regions of the eastern

Weddell Sea (70°E) to less than 2° in the Ross Sea near 170°E. Nevertheless, those differences are much smaller than the variations of the frontal positions themselves as a function of longitude, explaining the high correlation between the ice edge position and the one of the fronts.

The main element that explains these changes in the distance between the winter ice edge and the fronts is the variation in the latitude of the fronts and of the ice edge themselves: the further north the fronts are, the larger the distance between the fronts and the winter ice edge. The correlation between the distance from the fronts to the ice edge and the latitude of the fronts at the same longitude is higher than 0.80 (i.e. slightly less than the correlation between the latitude of the fronts and the latitude of the ice edge), except for the SBdy following the definition of Orsi et al. (1995) (Table 2). Part of this high correlation may be due to rapid changes in the position of some fronts, such as around 220°E or 330°E, which are weaker or absent in the ice edge latitude (Fig. 1). Indeed, by construction, these rapid changes are also present in the distance between the front and the ice edge and could increase the correlation between the distance from the fronts and the ice edge with the latitude of the fronts. However, the value is still higher than 0.50 for the PF and the sACCF if we make the correlation between the distance from the fronts to the ice edge and the latitude of the ice edge, for which these rapid changes in front positions, absent from the ice edge location, would tend to decrease the correlation (Table 2).

To interpret the correlations between the ice edge—front distance and the latitude of the fronts, we focus on the PF because of its stronger surface signal. Following a general principle, in winter, the heat transported from the latitude of the PF to the ice edge must balance at all longitudes a significant fraction of the oceanic heat loss to the atmosphere in order to keep surface temperatures above freezing and restrain sea ice advance. At southernmost ice edge latitudes, as in the Pacific sector around 260°E, atmospheric temperatures are colder, oceanic heat loss is stronger, and thus a larger heat flux from the PF is required to prevent ice formation. Since transport decreases with distance because of this surface heat loss, this is only possible over relatively short distances, forcing the ice edge to remain close to the PF. By contrast, when the ice edge is further north, the atmosphere is milder, oceanic heat loss is smaller, and less heat is needed to prevent freezing. In this case, the PF can be more distant from the ice edge, while still providing enough heat to prevent ice formation.

It is also possible that where southerly winds (i.e. winds from the south, corresponding to positive values on Fig. 4) tend to favour sea ice expansion by pushing it northward toward the PF, the frontal influence on the ice edge might be stronger, inducing a smaller distance between the front and the ice edge. However, this interpretation is not valid, as the opposite is observed at large scales, with on average stronger southerly winds (such as in the Weddell Sea between 300° and 360°E) associated with a larger distance between the PF and the winter ice edge (Fig. 4). This leads to a positive correlation between the meridional wind velocity at 60°S and the distance between the PF and the ice edge (Table 2). We have chosen 60°S here as it is close to the mean position of the winter ice edge (Fig. 3) but the results are not sensitive to a change of a few degrees of this latitude or if we chose to use instead the winds at the latitude of the winter sea ice edge. The positive correlation does not mean that sea ice transport has no impact on the ice edge location as it can be a dominant term in the sea ice mass balance there. However, this is consistent with the general view that sea ice advance during autumn and winter is more controlled by thermodynamic processes than by the wind-driven transport of sea ice to ice-free regions (Himmich et al., 2023; Goosse et al., 2023). Furthermore, winds can have an influence on the link between the front and ice edge positions through alternative ways as discussed below (subsection 3.d).

c/ Oceanic processes responsible for the link between the position of the ACC and the winter ice edge

A distance of several degrees between the PF and the ice edge in many areas implies that the PF does not limit sea ice expansion simply by inducing melting of the ice that would cross it, or by preventing ice formation in the warm waters on the front's northern flank. The next step is now to determine the mechanisms underpinning the high correlation between ice edge latitude and PF latitude found above (Table 2). By construction, ocean surface isotherms follow a path roughly parallel to the PF. This is also valid for the sea ice edge, which corresponds to the SST at freezing point and thus the -1.8°C isotherm. Surface waters just south of the PF are thus much warmer than at higher latitudes, and mean currents south of the PF contribute to a part of the southward heat transport from the PF to the location of the ice edge. The positions of the sACCF and SBdy, specifically following the definition of Park et al. (2019) based on dynamic topography, give indications on the direction of those heat-carrying currents. The sACCF and SBdy are close to the PF around 20°E and 220°E, and exhibit a southward translation east of those longitudes (Fig. 3). This is associated with a southward transport of warmer waters and a southward deflection of the ice edge.

The development of the subpolar gyres (which are located southward of the ACC) is connected to the path of the ACC itself, with both the gyres and the ACC being controlled by the oceanic bathymetry (Armitage et al., 2018; Patmore et al., 2019; Wilson et al., 2022). The southward shifts of the fronts and associated large-scale currents at 20°E and 220°E correspond to the traditional eastward limits of the Weddell and Ross gyres (e.g. Dotto et al., 2018; Vernet et al., 2019). Each of these regions is located in a southward branch of a gyre, contributing to the oceanic heat transport towards the Antarctic continent. While this transport is essential for the heat balance at high latitudes, the Weddell and Ross gyres are located to the south of the winter ice edge. Consequently, the gyres do not directly transport heat to the region of the winter ice edge. However, they can play an indirect role through their impact at higher latitudes and on the sea ice advance in fall, when the ice edge is positioned to the south of the gyres' northern limit (Morioka and Behera, 2021; Neme et al., 2021). Furthermore, the subpolar circulation cannot be reduced to the Weddell and Ross gyres (e.g., Sonnewald et al., 2023). In particular, it has been argued that smaller sub-gyres are present in other sectors of the Southern Ocean, as in the Indian Sector (Aoki et al., 2010; Yamazaki et al., 2020), where they could be closer to the ice edge and contribute to meridional exchanges there.

It is commonly thought that the ACC fronts limit meridional exchanges across the Southern Ocean. However, downstream of the main topographic obstacles, the mesoscale eddy formation is enhanced and the fronts generally meander and separate into finer frontal filaments, inducing enhanced meridional transport compared to other regions (Sallée et al., 2011; Thompson and Sallée, 2012; Dufour et al., 2015; Chapman, 2017; Rintoul, 2018; Denes et al., 2025). Consistently, the Eddy Kinetic Energy (EKE) peaks after Drake Passage at 300°E, around 40°E close to Crozet Plateau, at 90°E east of the Kerguelen Plateau, at 160°E and around 220°E (Fig. 5a). As the mesoscale eddies transport heat southward, each of these regions is associated with large vertical heat fluxes from the ocean to the sea ice in the NEMO simulation (Fig. 5b). This heat flux induces melting and limits sea ice formation, with a clear local impact in some regions as illustrated by a southward shift of the winter ice edge close to some of those peaks in EKE – such as at 40°E, 90°E and 220°E (Fig. 6). Besides, the peak in EKE at 160°E is not associated with a southward shift of the winter ice edge. At this longitude, the ice edge is already at high latitudes compared to other sectors, and the winter ice edge follows very closely the path of the mean ACC flow, as indicated by the location of the fronts (see Fig. 1). The direct impact of the eddies generated north of the ice edge may thus have a weaker relative effect there. The situation also becomes especially complex close to the Drake Passage (300°E), where eddy activity is prominent but the dynamics of the ice edge is influenced by many local processes. In addition to this influence of the eddy activity on the winter sea ice edge close to some hotspots, eddies might also have an impact in other regions, with for instance a larger background EKE between 60°E and 170°E where the ice edge

is located more southward and a smaller value of EKE between 300°E and 360°E where the ice edge is more northward. However, the large-scale connection between EKE and the ice edge (Fig. 6) appears 389 weaker than between the ice edge and the front position (Fig. 3).

It can also be seen on Figure 5b that the oceanic heat flux at the ice base does not show a clear distinction between the regions where the sACCF and SBdy are southward of the ice edge compared to the regions where it is northward of the ice edge (see Fig. 2). This is in agreement with the hypothesis proposed above that the signature of the sACCF and SBdy on subsurface temperatures does not necessarily lead to a strong signal at the ocean surface and thus on the position of the sea ice edge.

395 d/ Atmospheric processes responsible for the link between the position of the ACC and the winter ice 396 edge

The position of the fronts also influences atmospheric temperature, with the PF being located in the vicinity of the 2° or 3°C isotherms of surface air temperature in September, while the ice edge is associated with the -2 or -3°C isotherms (Fig. 7a). A northern position of the PF and a larger expansion of Antarctic Surface Waters in some sectors, like in the Weddell Sea, are thus related to lower air temperatures compared to the same latitude at longitudes where the PF is shifted further southward, like in the Amundsen Sea. This colder air contributes then to maintaining a large sea ice extent in regions where the PF is more northward. When analysing in greater detail Figure 7, the isotherms of surface air temperature appear more zonal than the fronts, with the 3°C isotherm being closer to the PF when this front is at lower latitudes, but further north than the PF when the front is at higher latitudes, in particular in the Bellingshausen and Amundsen seas.

In winter, the ocean tends to warm the atmosphere at high latitudes. Where the PF is located further south, sea surface temperatures are higher (relative to the atmosphere above), leading to a larger ocean-atmosphere temperature difference and thus a stronger oceanic heat loss. This oceanic heat warms the atmosphere, which transports part of the thermal energy southward, maintaining warmer air temperatures there and a more southward ice edge in the sector. The oceanic heat flux at the surface is influenced by many processes, including feedbacks between the atmosphere, the ocean and the sea ice, which complicates the identification of the causes of the changes. The dominant signal (Fig. 7b) is clearly the strong reduction of oceanic heat loss in ice-covered regions compared to ice-free areas. Nevertheless, the oceanic heat loss is much larger and on a wider latitude band in the Amundsen-Bellingshausen Sea, where the PF is displaced further southward than in other regions, consistent with the interpretation above.

The direction of the winds also strongly influences the transport of the heat extracted from the ocean north of the PF, toward the south where it could hamper sea ice formation or induce sea ice melting. Consequently, the southward atmospheric heat transport is larger, leading to a smaller distance between the PF and the ice edge, where the winds are stronger in the southward direction like in the Bellingshausen Sea than where they are more northward like in the Weddell Sea. This leads to a positive correlation between the winds at 60°S and the distance between the ice edge and the PF (Table 2, Fig. 4). Stronger northward winds could push the sea ice to the north, closer to the fronts. This would lead to a negative correlation between the meridional winds and the distance between the ice edge and the PF (see section 3b). However, this effect is overcompensated by the influence of the meridional winds via the heat transport, which explains the positive correlation. It should also be mentioned that the correlations decrease if we use a latitude of 50°S or that of the PF instead of 60°S. This suggests that the latitude of the ice edge is influenced more by the heat that reaches the highest latitudes than by the transport at the latitude of the PF itself.

However, the links between winds, fronts and sea ice edge position imply mutual interactions that complicate the interpretations of simple correlations. First, although the response of the ACC to winds and its variations is not simple and buoyancy forcing may also contribute to the ACC transport, winds are a key element in the ACC dynamics (Rintoul et al., 2001; Meredith and Hogg, 2006; Klocker et al., 2023b). Second, meridional winds influence atmospheric heat transport in all sectors and several studies have argued on the dominant influence of the strength of the meridional winds on the position of the ice edge even without discussing explicitly the potential role of the ACC or of the fronts (e.g., Bitz et al., 2005; Raphael, 2007; Raphael and Hobbs, 2014). This is consistent with the correlation between the position of the winter ice edge and climatological meridional winds at 60°S in September that reaches 0.76. Third, the bathymetry, which controls the position of the fronts, is linked to the topography of the continents, such as the location and height of the Antarctic Peninsula, which also influence wind and thereby provide an additional association between sea ice, winds and frontal positions.

By contrast, although westerly winds have a large impact on the oceanic upwelling and thus on the temperature at depth in the ocean, the correlation between the location of the Antarctic Divergence and that of the winter ice edge reaches only 0.43, i.e. much less than between the position of the fronts and that of the ice edge. The Antarctic Divergence is defined here as the latitude at which the climatological zonal mean wind velocity is equal to 1 m s<sup>-1</sup> in September. We have chosen this value instead of the traditional definition, based on a zero mean zonal wind, as in some regions of the Ross Sea the value is positive at all the latitudes of the Southern Ocean in the ERA5 reanalysis. This avoids the occurrence of undefined values for some longitudes and induces only a very minor shift in the other regions.

### e/ A simple regression model

In order to summarize the various contributions discussed above, we have developed a very simple linear regression model predicting the position of the winter ice edge as a function of key variables (Fig. 8). The first selected variable is the latitude of the Polar Front. We have chosen the PF compared to the other fronts because it has a more consistent link with the position of the ice edge. It is also more independent of sea ice processes than the sACCF and SBdy and thus limits the risk of a circular reasoning. The second variable is the mean eddy kinetic energy as a measure of the potential effect of eddy-induced heat transfer towards the ice edge. In contrast to the latitude of the ice edge, the EKE could vary by one order of magnitude between nearby regions (Fig. 5a), which could reduce the skill of a linear regression model. We have thus used here the logarithm of the EKE. The third variable is the magnitude of the meridional wind at 60°S, to account for the contribution of the atmospheric circulation. We have not selected atmospheric surface temperature or sea surface temperature because the feedbacks with sea ice are too strong.

A reference model assuming a constant latitude of the winter ice edge (60.6°S) leads to a Root Mean Square Error (RMSE) of 3.06° (Table 3). When using only one of the three variables at a time in the regression model, the agreement with the observed position of the ice edge is already good for the PF latitude (definition of Park et al., 2015) with an error of 1.28° of latitude (Table 3). The RMSE is higher for the meridional wind magnitude (1.98°) and the EKE (2.63°). When combining the different variables in multiple regressions, the improvement compared to the model using only the PF latitude is small with a minimum of 1.14° when the three variables are applied. The improvements come only from the EKE, with very limited contribution from the winds at 60°S. This suggests that the EKE is not the best predictor of the ice edge position (largest RMSE when used alone) but it brings some complementary information compared to the position of the PF. By contrast, including the wind provides only very limited added value in the multiple regression models, likely because of all the connections between

winds, ice edge and front position mentioned above. Despite those connections, the much better skill of the regression model using the PF latitude than the one using winds indicates a more direct influence of the location of the winter ice edge from the former than the later. Conclusions are similar when using the definition of the front of Orsi et al. (1995) (Table 3), except that the improvement brought by adding EKE and winds are even smaller.

**Table 3.** Root mean square errors (RMSE) of regression models predicting the latitude of the ice edge using as predictors the latitude of the Polar Front (PF), the mean of the logarithm of the Eddy Kinetic Energy averaged between 45°S and the continent (EKE) and the mean 10m wind velocity in September at 60°S (Wind). The RMSE corresponding to a constant latitude of the ice edge is also included as a reference.

| 1 | o | 7 |
|---|---|---|
| + | o | / |

| Predictors        | RMSE (°) |      |  |
|-------------------|----------|------|--|
|                   | Park     | Orsi |  |
| Constant latitude | 3.06     |      |  |
| PF                | 1.28     | 1.33 |  |
| EKE               | 2.63     |      |  |
| Wind              | 1.98     |      |  |
| PF and EKE        | 1.15     | 1.27 |  |
| PF and Winds      | 1.19     | 1.31 |  |
| PF, EKE and Winds | 1.14     | 1.26 |  |

#### 

### 4 Conclusions

The dynamics of the ACC and its interactions with the Antarctic sea ice cover involve a wide range of processes, including modulation of meridional oceanic heat transports by the mean ACC flow and mesoscale eddies, vertical fluxes between the surface ocean mixed layer and the oceanic interior, as well as exchanges and feedbacks with the atmosphere. Examining in detail all of these processes and quantifying their individual contributions to the climatological position of the winter sea ice edge would require sophisticated analyses and dedicated experiments (such as those of Wei and Chen, 2025 to analyse the zonal SST variations in the Southern Ocean). Nevertheless, we have shown here that simple diagnostics are sufficient to provide a physical understanding and to identify the first-order processes responsible for the strong influence that the ACC visibly exerts on the ice edge. This simple approach gives a broad baseline picture of the behaviour of the system, and provides a clear and intuitive interpretation compared to more complex analyses, especially for non-specialists of the subject (see Fig. 9 for a summary of the main processes).

We were able to confirm and quantify the standard hypothesis that the position of the winter sea ice edge is strongly linked to the position of the ACC fronts. This link is not only evident in some sectors where the fronts align with the ice edge, as identified in earlier studies (e.g., Roach and Speer, 2019; Ferola et al., 2023), but more generally across all regions. As the ACC path is itself largely controlled by the bathymetry of the Southern Ocean, the position of the winter ice edge is thus also mainly controlled by the bathymetry. The correlation with the winter ice edge is quantitatively similar for all the ACC fronts, with values exceeding 0.85. This similarity is expected, as the fronts, defined following standard criteria, remain roughly parallel to each other around Antarctica. Nevertheless, the link with the ice edge location is most consistent for the Polar Front. The winter sea ice edge always remains south of the Polar Front, while it occasionally lies north or south of the southern ACC front and of the Southern Boundary of the ACC. The sACCF and SBdy characterize shifts in the properties of warm

waters at depth. These warm waters play a central role in the Southern Ocean, and also have an impact on sea ice freezing and melting, but their impact on sea ice first requires that they are entrained into the mixed layer. Entrainment could occur in areas distant from the fronts and the ice edge, modifying surface ocean temperatures not just locally but at larger scales, primarily affecting the sea ice volume inside the pack rather than sea ice extent (Martinson et al., 1990). Depending on the strength of this entrainment, the sea ice edge is very close to the location of the sACCF and SBdy in some regions, but, in other areas, the connection with the ice edge is weaker than with the PF

Although the Polar Front influences the ice edge at large scales, the relationship can be either relatively strong or quite loose, with the PF being located from 2 to 10 degrees north of the mean winter ice edge. The distance is smaller when the front is located farther south, consistent with the idea that a front located farther south has a stronger influence on the sea ice edge. Indeed, a PF located farther south increases oceanic surface temperatures, thereby limiting sea ice formation at high latitudes where it would normally occur. Nevertheless, as the mean winter ice edge never reaches the PF, the heat energy available at the PF must be transferred southward to melt sea ice or reduce freezing. A first contribution is associated with large-scale ocean currents and eddies. The fronts are not hermetic barriers and some cross-frontal oceanic transport occurs, in particular downstream of the main bathymetric obstacles where mesoscale eddy activity is large. This implies locally enhanced ocean-ice heat fluxes and southward shifts of the ice edge. Around 20°E and 160°E, enhanced eddy activity occurs in regions of mean southward currents, also influenced by topography and contributing to the southward heat transport. Second, the high sea surface temperatures observed at high latitudes in regions where the PF is displaced southward act to warm the atmosphere above. The atmosphere transports then part of this heat toward the south, maintaining warmer air conditions that limit sea ice expansion. This atmospheric heat transport is likely favoured in regions where the winds are directed more southward, i.e. from the PF towards the ice edge. In this way, the heat is sourced from the ocean at the PF, but the atmosphere mediates the last stage of transport toward ice-covered regions.

Because of the strong links between some variables, it has not been possible, with the observation-based approach used here, to disentangle their respective role in setting the position of the winter ice edge. In particular, it has not been not possible to quantify the relative contribution of the mean currents to changes in the ice edge position compared to the influence of mesoscale eddies. We were also not able to make a clear distinction between a direct influence of the winds on the ice edge location compared to an indirect one through their role in the southward transport of the heat released at the PF location. It would be interesting to perform some dedicated sensitivity experiments with climate models, modifying either the sea surface temperature, winds or the bathymetry (and the thus the position of the ACC) to investigate in more details how each of these elements individually and in conjunction influences the positions of the ice edge.

We hope that the simple framework and conclusions reached here fill a gap between studies focused on the ACC dynamics and investigations of the processes responsible for the sea ice advance. We also hope that our work will stimulate research on the links between the ACC and sea ice variability, as well as their response to climate change. In this framework, the interaction with the ACC could first reduce the sea ice variability where the fronts are close to the ice edge, by limiting strongly the northward expansion of the pack and capping sea ice variations. Secondly, the variability of frontal positions could underpin changes in sea ice extent. It would be very instructive to investigate how these two potentially opposed ACC contributions to sea ice variability interact, and which one dominates as a function of location and timescale. We may also speculate that, in a warming world, if the latitude of the fronts remains stable because of the control by bathymetry, the distance between the PF and ice edge will increase because of the expected sea ice retreat and then the influence of the PF on the

559 position of the ice edge might decrease. In contrast, in a colder climate, the sea ice could be pushed 560 closer to the PF which would potentially limit directly the sea ice expansion. However, it is also possible 561 that the position of the PF would change or that its characteristics in cold conditions would become 562 closer to those of the current SBdy and sACCF with sea ice present both north and south of the PF. 563 Those questions are important to understand the variations of the sea ice cover but more generally 564 the global climate. For instance, carbon dynamics in the Southern Ocean, which are considered as a 565 main driver in glacial-interglacial transitions, are also strongly influenced by the front position and the 566 winter sea ice extent (e.g., Martinson, 2012; Skinner et al., 2010; Sigman et al., 2021; Ai et al., 2024).

### Data and code availability.

582

595

600

**OSI-SAF** sea ice concentration data can be downloaded at https://osi-569 saf.eumetsat.int/products/osi-450 (1979-2015) and https://osi-saf.eumetsat.int/products/osi-430-b-570 complementing-osi-450 (2015 onwards). The ERA5 data is available from the Copernicus Climate 571 https://cds.climate.copernicus.eu/cdsapp#!/dataset/reanalysis-era5-single-levels-monthly-572 means?tab=overview (Hersbach et al. 2020b) . NEMO and XIOS (a NEMO-compatible I/O library) are 573 developed by the NEMO consortium (https://www.nemo-ocean.eu/) and distributed under the CeCILL 574 license (http://cecill.info/licences/Licence CeCILL V2-en.txt). The ocean climatology of Yamazaki et al. 575 (2025) is available via Zenodo (https://zenodo.org/doi/10.5281/zenodo.12697777). The bathymetric 576 data has been taken from ETOP1 (NOAA National Geophysical Data Center. 2009, available at 577 https://www.ncei.noaa.gov/access/metadata/landing-

page/bin/iso?id=gov.noaa.ngdc.mgg.dem:316). EKE Dataset is available on SEANOE with the <a href="https://doi.org/10.17882/81032">https://doi.org/10.17882/81032</a> (Auger et al. 2021). The latitude of the fronts has been downloaded from <a href="https://australianantarcticdivision.github.io/orsifronts/articles/orsifronts.html#orsi-fronts-for-r">https://australianantarcticdivision.github.io/orsifronts/articles/orsifronts.html#orsi-fronts-for-r</a> (last Access 13-12-2024).

#### Acknowledgments

Hugues Goosse is a Research Director within the F.R.S.-FNRS (Belgium). This work was performed in 584 the framework of the PDR project T.0101.22 "Role of WINDS and oceanic interactions on Sea ice 585 Changes in the SOuthern Ocean over the Past millennium (WindSCOOP)" of the FRS-FNRS. We would 586 like to thank Bianca Mezzina, Augustin Lambotte and Antoine Barthélemy for help with the dataset 587 and the figures. The computational resources were provided by the Center for High Performance 588 Computing and Mass Storage (CISM) of the Université catholique de Louvain (UCLouvain) and the 589 Consortium des Équipements de Calcul Intensif en Fédération Wallonie-Bruxelles (CÉCI), funded by the 590 Fonds de la Recherche Scientifique de Belgique (F.R.S.-FNRS) under convention 2.5020.11 and by the 591 Walloon Region. Alberto Naveira Garabato acknowledges U.K. Research and Innovation guarantee 592 funding for a European Research Council Advanced Grant (EP/X025136/1). Benjamin Richaud is 593 supported by the Belgian Science Policy Office (BELSPO) under the RESIST project (contract no. 594 RT/23/RESIST).

### **Author contributions**

HG initiated the study and designed the analyses after discussions with all the co-authors. HG and BR performed the diagnostics. HG, BR and SL drew the figures. All the co-authors contributed in the interpretation of the results. HG wrote the initial version of the manuscript and the revisions, with inputs from all co-authors.

#### Competing interests.

The contact author has declared that none of the authors has any competing interests.

# 603 Figures

Figure 1. Locations of the four main ACC fronts: the Polar Front (PF), the Southern ACC front (sACCF), the southern boundary of the ACC (SBdy) and the Subantarctic Front (SAF) following definitions of (a) Park et al. (2019) and (b) Orsi et al. (1995) and of the climatological mean position (1979-2023) of the September ice edge (Lavergne et al. 2019). Bathymetry is shown in background (in m, Amante and Eakins, 2009). Name of key bathymetric features have been added on panel b.

Figure 2. Longitudinal oceanographic sections of climatological September temperatures (in °C) (Yamazaki et al., 2025) for two longitudes where the SBdy and sACCF are south of the ice edge (70 and 260°E) and two longitudes where the SBdy and sACCF are north of the ice edge (20 and 216°E). The position of the location of the winter ice edge is also included. The definition of the fronts follows the definition of Orsi et al. (1995)

Figure 3. Latitude of the fronts and of the climatological mean winter ice edge as a function of longitude, using definitions of (a) Park et al. (2019) and (b) Orsi et al. (1995).

Figure 4. Distance (in degrees of latitude) between fronts and the winter ice edge using definitions of (a) Park et al. (2019) and (b) Orsi et al. (1995). The climatological meridional 10m wind velocity in September at 60°S has been added (dotted orange line, in m/s).

Figure 5. (a) Eddy Kinetic Energy ( $m^2s^{-2}$ ) derived from satellite observations (Auger et al. 2022 ) with the location of the Polar Front following the definition of Park et al. (2019) and of the climatological mean position of the September ice edge. (b) Sensible oceanic heat flux at the sea ice base in September (in W  $m^{-2}$ , averaged over 1979-2023) simulated by NEMO.

Figure 6. Mean of the Eddy Kinetic Energy (m<sup>2</sup>s<sup>-2</sup>) averaged between 45°S and the continent with the latitude of the Polar Front following the definition of Park et al. (2019) and the latitude of the sea ice edge in September.

Figure 7. (a) September mean (1979-2023) air temperature at 2m (in K) from ERA5 (Hersbach et al., 2020a). (b) Heat flux at the ocean surface (in W m<sup>-2</sup>, positive downward) avaraged over the years 1979-2023 for the months August in the simulation performed with the model NEMO. In contrast to other figures for which September conditions are shown, we have chosen August for this figure as heat losses reach their maximum earlier than the sea ice extent. The location of the PF (Park et al., 2019) and the winter sea ice edge have been added.

Figure 8. Latitude of the observed ice edge compared to the one predicted by regression models using as predictor the latitude of the polar front (R. PF), the mean of the logarithm of the eddy kinetic energy southward of 45°S (R. EKE), the 10m wind velocity at 60°S in September (R. Winds) and the multiple linear regression using those 3 variables together (Reg3).

Figure 9. Figure summarizing the main mechanisms linking the position of the Polar Front and that of the winter sea ice edge.

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
