# Peer review of "Winter sea ice edge shaped by Antarctic Circumpolar Current"

_EGUsphere, 2025_

## Referee Comment (RC2)

**Manuscript "On the control of the 1 position of the winter sea ice edge by the Antarctic Circumpolar Current."**
Authors: Hugues Goosse, Stephy Libera, Alberto C. Naveira Garabato, Benjamin Richaud, Alessandro Silvano, Martin Vancoppenolle

The study discusses the influence of the Antarctic Circumpolar Current (ACC) on the position and dynamics of the winter sea ice edge in the Southern Ocean, highlighting the role of the Polar Front (PF) among various oceanic fronts and different heat transport mechanisms.
The definitions of ACC fronts are based on both oceanographic observations and satellite data, which are used to analyze their relationship with the mean location of the sea ice edge. The PF is identified as the best indicator of the ACC's influence on sea ice, due to the lower uncertainty in its position derived from observations and its greater independence from sea ice processes.
It influences sea ice advance and the position of the winter ice edge by modulating heat transport toward high latitudes, both through oceanic processes (via eddy-induced transport) and atmospheric processes (via meridional wind transport).
Through regression models predicting the winter ice edge position using the latitude of the PF, eddy kinetic energy (EKE), and wind velocity, the authors find that PF latitude is the most reliable predictor. Including EKE or winds leads to little or no improvement in predictive skill.

The manuscript is overall well written and clear. The methodology and results are presented in a clear and sufficiently detailed manner.
My only main comment concerns the use of mean wind speed at 60°S, which is often not co-located with the PF or the sea ice edge. This could result in only a partial assessment of the impact of winds on the atmospheric heat transport from the PF to the sea ice.

**Minor comments**

Line 37. "the sea ice seasonal cycle of sea ice cover" - there is a repetition

Figure 1,3,4. Blue and purple are not an ideal combination for color-blind safe figures. I suggest using a different color combination

Line 71. variability and noise are weaker than at surface

Line 70-81. It is not clear whether the two identification methods are linked or represent two distinct approaches. The paragraph could be rephrased to clarify this point.

Line 210. The PF's position relative to the ice edge is more consistent between regions. However without further evidence, you cannot conclude yet that PF's influence is more robust.

Line 211. The sACCF and the SBdy are close to the winter ice edge in several regions

Line 229-230. "which is systematically located…"- this phrasing is somewhat repetitive

Figure 2. Square brackets denote "units of" in scientific notation: [Depth] m, so it would be more correct to use (m) instead of [m]

Line 251-254. This sentence is not clear, please consider rephrasing for clarity

Line 263-266. The statement may not be fully supported unless the correlation with the distance from the coast or with other factors are also not reported.

Line 291-293. I think the fact that the distance between the front position and the sea ice edge shows a high correlation with latitude better explains the high correlation between front position and sea ice edge latitude, rather than the magnitude of this distance, as mentioned earlier (lines 286–288).

Line 315. It should probably be "stronger northerly winds." It may also be worth adding a clarification about the sign convention used for the wind direction.

Line 377. more zonal than the fronts

Line 388-390. It should be noted that there is also substantial heat loss in the Indian Ocean sector, despite the PF being located farther north.

Line 391-392. The phrase "the heat extracted from the ocean to the north of the PF, towards the south" is unclear, it should probably be "the heat extracted from the ocean north of the PF, towards the south"

Lines 393–395. Does the fact that in sectors where the PF is farther north it is close to 50°S—and thus the 60°S meridional winds may not be representative—affect the interpretation of the relationship with atmospheric heat transport?

Line 428. Similar to lines 393–395, it might be worth considering a latitude that varies with the PF's position, to better assess the impact of winds.

Line 501-505. This sentence is too long, please split it to make it more clear.

Line 511. it is also possible that the position

Line 513-516. This sentence is not clear, please consider rephrasing for clarity

Best Regards,
Martina Zapponini

---

## Author Comment (AC1)

**Response to the comments of Martina Zapponini**

The study discusses the influence of the Antarctic Circumpolar Current (ACC) on the position and dynamics of the winter sea ice edge in the Southern Ocean, highlighting the role of the Polar Front (PF) among various oceanic fronts and different heat transport mechanisms. The definitions of ACC fronts are based on both oceanographic observations and satellite data, which are used to analyze their relationship with the mean location of the sea ice edge. The PF is identified as the best indicator of the ACC's influence on sea ice, due to the lower uncertainty in its position derived from observations and its greater independence from sea ice processes. It influences sea ice advance and the position of the winter ice edge by modulating heat transport toward high latitudes, both through oceanic processes (via eddy-induced transport) and atmospheric processes (via meridional wind transport). Through regression models predicting the winter ice edge position using the latitude of the PF, eddy kinetic energy (EKE), and wind velocity, the authors find that PF latitude is the most reliable predictor. Including EKE or winds leads to little or no improvement in predictive skill.

The manuscript is overall well written and clear. The methodology and results are presented in a clear and sufficiently detailed manner.

We would like to thank the reviewer for the positive evaluation and the suggestions that will contribute to improve the quality of our manuscript. Our responses below are in blue and the suggested modifications in the text in green.

My only main comment concerns the use of mean wind speed at 60°S, which is often not co-located with the PF or the sea ice edge. This could result in only a partial assessment of the impact of winds on the atmospheric heat transport from the PF to the sea ice.

We selected the mean wind speed at 60°S as it is representative of the winds at high latitudes in the Southern Ocean, close to the ice edge. Compared to the winds at the latitude of the PF or of the sea ice edge, this choice has the advantage that it does not require any information on the location of the PF or of the ice edge and is thus independent of those locations. As suggested by the reviewer, we have repeated the analyses using the wind speed at the location of the PF and of the ice edge. This has some influence on the values obtained but without major changes in our conclusions. This is discussed in more details below in the response to the specific comments on this point.

Minor comments

Line 37. "the sea ice seasonal cycle of sea ice cover" - there is a repetition

This will be removed in the revised version.

Figure 1,3,4. Blue and purple are not an ideal combination for color-blind safe figures. I suggest using a different color combination

Before submission, we have checked the figures using the site https://www.color-blindness.com/coblis-color-blindness-simulator/ to choose the colors. There is no ideal solution but we find that the selected colors were a good compromise, as they have different lightness values. We also tested the option to have dashed or dotted lines but the front positions display rapid fluctuations and this breaks in some region the continuity of the curve. We are of course open to all the suggestions to improve the quality and readability of our figures and happy to change them at any stage, including at the production stage, if a better option is found.

Line 71. variability and noise are weaker than at surface

Thanks, this will be corrected in the revised version.

Line 70-81. It is not clear whether the two identifications methods are linked or represent two distinct approaches. The paragraph could be rephrased to clarify this point.

The two identifications methods are based on different observations (satellite versus in situ) and criteria. However, they are linked as the criteria selected to define the front from satellite observations are selected to match the position of the front inferred from in situ observation in key locations, where the fronts are well defined and observed.

After a few rounds of discussion among co-authors, we reached a consensus to reduce the technical/methodological information inside the introduction section and focus on that in the methods section. So, we will direct the readers to section 2a and include in section 2a the following.

In the introduction :
The correspondence between the sea surface height (SSH) gradients linked to those jets and specific SSH values have led to another method for identifying frontal positions using a judicious choice of circumpolar SSH contours. See section 2a for more details on frontal definitions.

In section 2a:
These two definitions are thus linked but they are based on independent data sets (i.e. in situ measurements of ocean properties for Orsi et al., 1995, and satellite observations of SSH for Park et al., 2019).

Line 210. The PF's position relative to the ice edge is more consistent between regions. However without further evidence, you cannot conclude yet that PF's influence is more robust.

We will suppress the word 'robust' as we agree that we cannot conclude without additional evidence.

Line 211. The sACCF and the SBdy are close to the winter ice edge in several regions

Thanks, this will be corrected in the revised version.

Line 229-230. "which is systematically located…"- this phrasing is somewhat repetitive

We propose to replace 'systematically' by 'always' but we consider that the part of the sentence after 'which' is necessary to have an explicit mention of the conditions for both the sACCF and the SBdy (close to the ice edge) and the PF (north of the ice edge). We could keep part of it implicit but we fear that it may be less clear for some readers, although we agree that for some other readers it may imply a repetition of the information.

Figure 2. Square brackets denote "units of" in scientific notation: [Depth] m, so it would be more correct to use (m) instead of [m]

As suggested, we will use (m) instead of [m] on Figure 2.

Line 251-254. This sentence is not clear, please consider rephrasing for clarity

The goal of this sentence was to insist that, while the presence of warm water at depth is not a strong criterion to determine the position of the winter sea ice edge, it can still play a large role for other elements of the system, such as the melting of ice shelves. We propose to rephrase it to make this clearer:

Besides, the presence of warm water at depth and its southward transport can have a large impact on other elements of the Southern Ocean system. For instance, a recent southward shift of the Circumpolar Deep Water, the sACCF and the SBdy has been observed off East Antarctica, potentially leading to additional melting of the ice shelves in this region (Yamazaki et al., 2021; Herraiz-Borreguero and Naveira Garabato, 2022).

Line 263-266. The statement may not be fully supported unless the correlation with the distance from the coast or with other factors are also not reported.

We agree with the Referee that this statement is not supported by our results. We propose thus to remove this sentence in the revised version. The alterative option to compute the correlation between the ice edge and other factors (such as the distance to the coast but potentially many others) could distract the reader from the flow of the discussion which is on the role of the ACC, not on other potential factors.

Line 291-293. I think the fact that the distance between the front position and the sea ice edge shows a high correlation with latitude better explains the high correlate on between front position and sea ice edge latitude, rather than the magnitude of this distance, as mentioned earlier (lines 286–288).

From our analyses, the correlation of the distance from the PF position to the sea ice edge with the latitude of the front reaches 0.82 and 0.83 for the two definitions of the front. The values are thus lower than for the correlation of the latitude of the front with the latitude of the ice edge (0.92 and 0.90). This will be specified in the revised version. However, those lines are mainly descriptive, discussing only the correlations, so we prefer not to go deeper in the potential interpretation at this stage. We will also change 'reason' to 'element that explains' to insist on this descriptive objective:

The main element that explains these changes in the distance between the winter ice edge and the fronts is the variation in the latitudes of the fronts and of the ice edge themselves: the further north the fronts are, the larger the distance between the fronts and the winter ice edge. The correlation between the distance from the fronts to the ice edge and the latitude of the fronts at the same longitude is higher than 0.80 (i.e. slightly less than the correlation between the latitude of the front and that of the ice edge), except for the SBdy following the definition of Orsi et al. (1995) (Table 2).

Line 315. It should probably be "stronger northerly winds." It may also be worth adding a clarification about the sign convention used for the wind direction.

We checked and the sentence seems correct to us with 'stronger southerly winds associated with a larger distance between the PF and the winter ice edge'. However, to avoid confusion we will add a definition of the convention in the first sentence of this paragraph:

It is also possible that where southerly winds (i.e. winds from the south, corresponding to positive values on Fig. 4) tend to favour sea ice cover expansion by pushing it northward toward the PF,…

Line 377. more zonal than the fronts

Thanks, this will be corrected in the revised version.

Line 388-390. It should be noted that there is also substantial heat loss in the Indian Ocean sector, despite the PF being located farther north.

As mentioned in the submitted version (lines 385-387), there are major losses in nearly all the regions close to the ice edge; including in the Indian Sector. Nevertheless, the region with stronger

oceanic heat loss than in any other sectors is in the Amundsen-Bellingshausen Sea. This will be specified more explicitly in the revised version :

Nevertheless, the oceanic heat loss is much larger and on a wider latitude band in the Amundsen-Bellingshausen Sea, where the PF is displaced further to the south than in other regions, consistent with the interpretation above.

The region in the Indian sector eastward of the Kerguelen Plateau is also characterized by large heat losses of a wide latitudinal range but on a narrower longitude band. We interpret this as consequence of the oceanic currents and strong eddy activity there. Consequently, we do not mention it in this section devoted to atmospheric processes to avoid confusion but discuss this point in the subsection devoted to oceanic transport.

Line 391-392. The phrase "the heat extracted from the ocean to the north of the PF, towards the south" is unclear, it should probably be "the heat extracted from the ocean north of the PF, towards the south"

Thanks, this will be corrected in the revised version.

Lines 393–395. Does the fact that in sectors where the PF is farther north it is close to 50°S—and thus the 60°S meridional winds may not be representative—affect the interpretation of the relationship with atmospheric heat transport?

We have repeated our analyses using the meridional winds at the locations of the PF and of the ice edge. The main message is that using the location of the ice edge instead of the value at 60°S does not change much our results. For instance, we have computed the correlation of the distance between the front and the ice edge with the meridional winds at the location of the ice edge and we obtained values of 0.68 for both the definitions of Orsi et al. (1995) and Park et al. (2019) (instead of 0.61 and 0.55, Table 2). This confirms our choice that using 60°S provides a good estimate of the value close to the ice edge, considering the consistency of climatological winds on scales of several hundreds of kms. By contrast, using values at lower latitudes gives lower correlations. For the correlation of the distance between the front and the ice edge with the meridional winds at either 50°S or the latitude of the PF, we obtain values that are always lower than 0.2. It suggests that the most important element is not the atmospheric heat transport at the PF itself but the conditions more southward that favor the transport of this heat to the region of the ice edge. For instance, we can imagine a case where the winds are southward at the PF but shift quickly to a northward direction between the PF and the ice edge. In that case, the southward atmospheric heat transport could be large at the PF but much lower close to the ice edge and thus have a smaller impact on it. We will add this additional information at two locations in the revised manuscript.

When we first use the latitude of 60°S, we will modify the text to

We have chosen 60°S here as it is close to the mean position of the winter ice edge (Fig. 3) but results are not sensitive to a change of this latitude by a few degrees, or to using instead the winds at the latitude of the winter sea ice edge.

Second, when we discuss the potential role of the atmospheric transport, we will add:

It should also be mentioned that the correlations decrease if we use a latitude of 50°S or that of the PF instead of 60°S. This suggests that the latitude of the ice edge is influenced more by the heat that reaches the highest latitudes than by the transport at the latitude of the PF itself.

Line 428. Similar to lines 393–395, it might be worth considering a latitude that varies with the PF's position, to better assess the impact of winds.

In the regression model, we selected variables that are as much independent of each other as possible. If we use the winds at the latitude of the PF, we will include information on both the winds and the position of the PF and disentangling the contributions of the winds from the one of the position of the PF would thus be more difficult. Furthermore, if we use the winds at the latitude of the PF, the correlations are much lower than for 60°S, and the RMS of the corresponding regression model higher.

Line 501-505. This sentence is too long, please split it to make it more clear.

We propose to remove in the revised version the parts of the sentence starting with 'which' that make the sentence longer without adding much information.

Line 511. it is also possible that the position

Thanks, this will be corrected in the revised version.

Line 513-516. This sentence is not clear, please consider rephrasing for clarity

To make this sentence clearer. We propose to split it in two sentences and to reformulate the second part:

Those questions are important to understand the variations of the sea ice cover, but more generally the global climate. For instance, carbon dynamics in the Southern Ocean, which are considered as a main driver of glacial-interglacial transitions, are also strongly influenced by the position of the fronts and the winter sea ice extent (e.g., Martinson, 2012; Skinner et al., 2010; Sigman et al., 2021; Ai et al., 2024).

---

## Author Comment (AC2)

**Response to the comments of Dr. Kaihe Yamazaki**

This manuscript investigates the influence of the ACC fronts on the climatological mean position of the Antarctic winter sea ice edge. Using established frontal definitions (Orsi et al., 1995; Park et al., 2019) and observational/reanalysis datasets for sea ice, atmospheric, and oceanic variables, the authors find strong correlations (> 0.85) between the latitudes of all major ACC fronts and the winter sea ice edge. The Polar Front (PF) is identified as the most consistent indicator. The study proposes two primary mechanisms for this control: 1) poleward heat transport by mesoscale eddies generated downstream of topographic barriers, and 2) atmospheric warming above warmer surface waters near the PF, with this heat subsequently transported poleward towards the ice, particularly with southward-directed winds. The authors conclude that bathymetry, by shaping the PF's path, strongly constrains the winter sea ice edge.

**General Comments:**

A very well-written, clearly structured, and valuable contribution to understanding the controls on Antarctic sea ice extent. The study addresses an important and under-explored link in a circumpolar manner. The use of multiple frontal definitions and a relatively simple yet effective methodology lends robustness to the main conclusions. The identified mechanisms are physically plausible and supported by the presented evidence and previous studies. The figures are generally clear and effectively support the text, with Figure 9 providing an excellent summary.

We would like to thank the reviewer for his positive evaluation of our manuscript and the constructive comments.

My concern is the relative lack of discussion on the role of subpolar gyres and the Antarctic Divergence. These features are intrinsically linked to the ACC, upwelling of CDW, and spread of WW, and thus might be highly relevant to SIE positioning. The central narrative that the "ACC/PF controls the winter sea ice edge," while supported by the presented correlations, might potentially be a trivialization or, at least, could benefit from a more nuanced discussion of these interconnected Southern Ocean dynamics.

We totally agree that the Southern Ocean dynamics includes many aspects that are interconnected and it is not easy to identify the contribution of individual mechanisms as they can be linked or driven by similar constraints, such as land topography or oceanic bathymetry. Specifically, we will extend in the revised version the discussion of the role of the subpolar gyres and more generally of the role of horizontal ocean currents. We hope this will bring a more nuanced interpretation of our results. We have also correlated the position of Antarctic Divergence with the one of the winter ice edge and found lower correlation than for the fronts. More details are given below following the related specific comments.

**Specific Comments:**

1. The manuscript does mention subpolar gyres (L338-340: "Consistent with the development of the subpolar gyres to the south of the ACC..."). However, their role feels somewhat secondary to the direct influence of the fronts. Subpolar gyres are major conduits for heat towards the Antarctic continent and potentially influence subsurface ocean heat content and sea ice formation/melt. I think the authors can

elaborate on how the ACC fronts interact with or shape these gyres, and how gyre dynamics themselves contribute to the SIE position. Is the gyre influence primarily a consequence of the ACC's path (as implied), or do they exert a more independent, synergistic control on the SIE alongside the fronts? Ultimately, the authors may want to present how the ACC is more important than gyres in locating the SIE. The discussion on EKE hotspots (L348-353) or somewhere around could be a place to better integrate gyre dynamics, as these are often associated with gyre boundaries or instabilities. As a consequence of such discussion, can we still say "the ACC is controlling the winter sea ice edge"?

We acknowledge that the discussion of the gyres and more generally of large-scale currents was brief in the submitted version. There were three main reasons for that.

- First, our outlook is that the path of the large-scale currents poleward of the ACC and the position of the gyres are all constrained by oceanic bathymetry. Therefore, the position of gyres and the position of the ACC, are directly connected by having the same constraint. In particular, the two of the main gyres (Weddell and Ross Gyres) form around the major embayments around the Antarctic continent and are shown to follow the major bathymetric features to the north. Because of those relationships, the position of the fronts (which we use as a central diagnostic in our study) brings some information on the ACC itself but also on the horizontal circulation and the location of the gyres. It is thus difficult to disentangle the direct contribution from the ACC from an indirect one coming for instance from the role of the gyres.
- Second, there are different views on the structure of the Subpolar gyres in the Southern Ocean with most studies mainly presenting only the two main gyres (Weddell and Ross Gyres) (e.g. Armitage et al. 2018; Dotto et al., 2018; Vernet et al., 2019), while some others present one supergyre from the Weddell Sea to the Ross Sea (e.g., Sonnewald et al., 2023), with also smaller subpolar gyres in the Indian Ocean sector (e.g., Yamazaki et al. 2020). In contrast to the other diagnostics presented in the submitted version, we did not find a good way to present a robust and circumpolar diagnostic allowing to study the link between the subpolar gyres and the position of the ice edge as for the other elements investigated here. Of course, this does not preclude the role of the gyres in some regions as we mentioned in the submitted version L338-340.
- Third, the Ross and the Weddell gyres are located south of the winter ice edge. Consequently, while they are essential elements of the heat balance at higher latitudes, they are not expected to contribute directly to the southward transport of heat to region of the winter ice edge. They can only influence it indirectly through their role on the advance of sea ice in fall when the ice edge is positioned southward of the northern limit of the gyre.

Nevertheless, we propose to expand the discussion of the potential role of the gyres replacing L338-340 of the submitted version by the following:

Being located south of the ACC, the development of the subpolar gyres is connected to the path of the ACC itself, with both the gyres and the ACC being controlled by the oceanic bathymetry (Armitage et al., 2018; Patmore et al., 2019; Wilson et al., 2022). The southward translation of the fronts and associated large-scale currents at 20°E and

220°E correspond to the traditional eastward limits of the Weddell and Ross gyres (e.g. Dotto et al., 2018; Vernet et al., 2019). In those regions, the circulation of each subpolar gyre is also southward, contributing to the oceanic heat transport towards the Antarctic continent. While this transport is essential for the heat balance at high latitudes, the Weddell and Ross gyres are located to the south of the winter ice edge. Consequently, the gyres do not directly transport heat to the region of the winter ice edge, but can play an indirect role through their impact at higher latitudes and on the sea ice advance in fall, when the ice edge is positioned to the south of the gyres' northern limit. Furthermore, the subpolar circulation cannot be reduced to the Weddell and Ross gyres (e.g., Sonnewald et al., 2023). In particular, it has been argued that smaller sub-gyres are present in other sectors of the Southern Ocean, as in the Indian Sector (Aoki et al., 2010; Yamazaki et al., 2020), where they could be closer to the ice edge and contribute to meridional exchanges there.

We will also ensure that the text is nuanced enough each time we discuss the role of the ocean in transporting heat southward, insisting that this transport is not limited to eddies but horizontal currents, in connection with the subpolar gyres, can also play a role.

We answer specifically the reviewer's point 'can we still say "the ACC is controlling the winter sea ice edge"?' in our answer to comment 3. below.

New references (compared to the submitted version)

Armitage, T. W. K., Kwok, R., Thompson, A. F., and Cunningham, G.: Dynamic topography and sea level anomalies of the Southern Ocean: Variability and teleconnections. J. Geophys. Res.:Oceans, 123, 613–630. https://doi.org/10.1002/2017JC013534, 2018

Aoki, S., Sasai, Y., Sasaki, H., Mitsudera, H., and Williams, G. D. (). The cyclonic circulation in the Australian–Antarctic basin simulated by an eddy-resolving general circulation model. Ocean Dynamics, 60(3), 743–757. https://doi.org/10.1007/s10236-009-0261-y, 2010

Patmore, R.D., Holland, P.R., Munday, D.R., Naveira Garabato, A.C., Stevens, D. P., and Meredith, M.P.: Topographic control of Southern Ocean gyres and the Antarctic Circumpolar Current: a barotropic perspective. J. Phys. Ocean. 49, 3221-3244. https://doi.org/10.1175/JPO-D-19-0083.1, 2019

Yamazaki, K., Aoki, S., Shimada, K., Kobayashi, T., and Kitade, Y. . Structure of the subpolar gyre in the Australian-Antarctic Basin derived from Argo floats. J. Geophys. Res.: Oceans, 125, e2019JC015406. https://doi.org/10.1029/2019JC015406, 2020

2. The Antarctic Divergence is a circumpolar feature characterized by the zero zonal wind and the associated surface Ekman upwelling. This potentially impacts mixed layer depth and temperature, which might be crucial for sea ice formation and the position of the SIE. I hope the manuscript should explicitly discuss the potential role of the Antarctic Divergence. How does its mean position relate to the ACC fronts and the SIE? Could variations in upwelling along the Divergence explain some of the regional variability in the SIE or the distance between the PF and the SIE? How is it

related to the surface meridional winds mentioned in the manuscript? The influence is not just about heat transported from the PF, but also about heat supplied from ocean closer to the ice edge via wind-driven divergence of ice floes.

The dynamics of the Southern Ocean and the characteristics of the water masses are strongly influenced by the Antarctic Divergence and the associated upwelling. However, the Antarctic Divergence is located southward of the winter ice edge at all longitudes (Figure R1). This means that the heat supplied to the surface at the Antarctic Divergence does not directly contribute to the balance at the ice edge but of course it can have an indirect effect, as discussed above for the gyres. Furthermore, the position of the Antarctic Divergence displays much less variation as a function of longitude than the winter ice edge or the Polar Front. The correlation between the Antarctic Divergence and the winter ice edge reaches 0.43 i.e. much less than between the position of the Polar Front and the one of the ice edge. This suggests that the constraint brought by the position of the Antarctic Divergence on the position of the winter ice edge is weaker than the one of the fronts. We have repeated the analysis with other diagnostics of the wind-driven divergence such as the latitude of the maximum wind stress curl or of the maximum of the derivative of the zonal wind stress as a function of latitude, arriving at similar conclusions. This is consistent with the broader picture that zonal winds (and atmospheric temperatures) display less zonal differences than oceanic variables such as the SST or the position of the fronts. This indicates that the differences in the latitudinal position of the ice edge between the different sectors is more controlled by oceanic processes (and the bathymetry) than by atmospheric ones such as the position of the zero zonal wind. This will be discussed in the revised version at the end of the new subsection 3d 'd/ Atmospheric processes responsible for the link between the position of the ACC and the winter ice edge':

By contrast, although westerly winds have a large impact on the oceanic upwelling and thus on the temperature at depth in the ocean, the correlation between the location of the Antarctic Divergence and that of the winter ice edge reaches only 0.43 i.e. much less than between the position of the fronts and that of the ice edge. The Antarctic Divergence is defined here as the latitude at which the climatological zonal mean wind velocity is equal to 1 m s$^{-1}$ in September. We have chosen this value instead of the traditional definition, based on a zero mean zonal wind, as in some regions of the Ross Sea the value is positive at all the latitudes of the Southern Ocean in the ERA5 reanalysis. This avoids the occurrence of undefined values for some longitudes and induces only a very minor shift in the other regions.

[Figure]

Figure R1. Latitudes of the climatological mean winter ice edge, of the Antarctic Divergence (Div) and of the Polar Front as a function of longitude, using definitions of Park et al. (2019) and Orsi et al. (1995). The Antarctic Divergence is defined here as the latitude at which the climatological zonal mean wind velocity is equal to 1 m s$^{-1}$ in September. We have chosen this value instead of the traditional definition, based on a zero mean zonal wind, as in some regions of the Ross Sea the value is positive at all the latitudes of the Southern Ocean in the ERA5 reanalysis. This avoids the occurrence of undefined values for some longitudes and induces only a very minor shift in the other regions.

3.  While the correlations are strong, the term "control" sounds like a very direct and dominant causal mechanism. The ACC fronts (and gyres) are themselves largely controlled by bathymetry. The paper argues the fronts are key mediators of this bathymetric influence on the SIE. This is plausible. While the authors do use "constrain" and "influence," consider if the overarching message of "control" is fully supported for all aspects, or if wording like "influence" or "constrain" could be more accurate in some contexts (especially when considering currently unaddressed roles of gyres and the Divergence).

    The word "control" may indeed seem too strong for some aspects. It is the reason why in many paragraphs of the text we used "constrain" and "influence". Our intention with the title 'On the control of the position of the winter sea ice edge by the Antarctic Circumpolar Current' was to state that the goal of the paper was to investigate this potential control, not to give the conclusion that such control is the only element setting the position of the ice edge. As this may give a wrong impression of our overarching message, we suggest removing the word 'control' to the title and to change it to 'Winter sea ice edge shaped by Antarctic Circumpolar Current pathways'.

    We consider that the word 'control' can still be used, for instance when we mention the influence of the bathymetry on the path of the ACC or the eddy hotspots, but we will check the whole manuscript for each occurrence of 'control' to ensure that it does not imply a dominant causal mechanism when this is not necessarily the case.

4.  The two proposed mechanisms (eddy heat transport and atmospheric heat transport mediated by SSTs near the PF) are The link between southward-directed winds and atmospheric heat transport (L391-396) is interesting. However, the correlation between meridional winds and the PF-SIE distance is positive (Table 2, Fig 4), suggesting stronger southerlies are associated with a larger PF-SIE distance. The text

(L312-318) acknowledges this and argues against wind-driven sea ice transport being the primary factor for this correlation. The subsequent argument for winds influencing atmospheric heat transport (L391-396) needs to be carefully reconciled with this earlier point to avoid reader confusion. Perhaps the argument is that despite stronger southerlies pushing ice north (which would intuitively decrease the PF-SIE distance if the PF were a fixed barrier), the atmospheric heat transport effect in regions with southward winds (from PF to SIE) is more dominant in setting the SIE further south (thus increasing the PF-SIE distance if the PF is far north). I think this needs very clear articulation.

The Reviewer is totally right, and that is the message we wanted to convey in this paragraph. The discussion was too short in the submitted version and we will reformulate the argument in the revised version to make this clearer, adding also a more explicit reference to the subsection where we discuss first the potential impact of meridional winds on sea ice transport:

This leads to a positive correlation between the winds at 60°S and the distance between the ice edge and the PF (Table 2, Fig. 4). Stronger northward winds could push the sea ice to the north, closer to the fronts. This would lead to a negative correlation between the meridional winds and the distance between the ice edge and the PF (see section 3b). However, this effect is overcompensated by the influence of the meridional winds on the heat transport, which explains the positive correlation.

5.  To better understand the spatial extent of the ACC front-SIE relationship, I hope the authors to consider estimating a characteristic horizontal length scale of the observed high correlations. Analyses like lagged spatial correlation or spectral methods could achieve this.

The correlation remains high on a large horizontal length scale (Figure R2), which is consistent with the latitude of the fronts and the ice edge that displays high auto-correlation on length scales of several tens of km. For instance, if we analyse the correlation between the latitude of the PF (definition of Orsi et al.,1995) and the position of the ice edge at different spatial lags, the correlation remains positive from lag of -100° of longitude (PF shifted westward) to a lag of 71° of longitude (PF shifted eastward). The correlation remains higher than 0.45 (half of the maximum value from a lag of -54° to 36°. The maximum correlation is found for a lag of -1° but the difference with the value at lag zero is very small (0.9005 compared to 0.8991). To highlight this point, we suggest adding in the revised version the following sentence when we discuss Fig.3:

The correlation remains high (with values larger than half of the peak correlation) for a spatial lag between the positions of the ice edge and of the front exceeding 30° of longitude, indicating that the observed high correlations have a large horizontal scale. The peak correlation is generally very close to a spatial lag of zero degrees of longitude.

[Figure]

Figure R2. Lagged spatial correlation between the latitude of the ice edge and the latitude of the Polar Front (following the definition of Orsi et al. 1995) for lags between -180° of longitude (PF shifted westward) and +180° of longitude (PF shifted eastward).

6. Although the focus on the climatological mean is a valid simplification, it would be beneficial to briefly acknowledge in the discussion that interannual variability of winter sea ice edge, regarding the recent sea ice extremes, even if it's beyond the scope of this paper.

As suggested we propose to add a short discussion of the interannual variability, as a perspective in the last paragraph of the revised version:

In this framework, the interaction with the ACC could reduce the sea ice variability where the fronts are close to the ice edge, by limiting strongly the northward expansion of the pack. Concurrently, the variability of frontal positions could underpin changes in sea ice extent. It would be very instructive to investigate how these two potentially opposed ACC contributions to sea ice variability interact, and which one dominates as a function of location and timescale.

7. L290: "the further north the fronts are, the larger the distance between the fronts and the winter ice edge." – Intriguing. I wonder why this is the case, and the authors might also want to explain more about it (perhaps in terms of the meridional gradient of ocean temperature and/or EKE).

In the submitted version, we discussed this positive correlation between the distance from the fronts to the ice edge and the latitude of the fronts in the paragraph following

the sentence (the paragraph including this sentence is only descriptive). We will add the information explicitly to make this clearer in the revised version. Our argument is that a Polar Front located more northward has a weaker relative contribution to the position of the ice edge than a front located more southward, where the climate is on average colder and the role of the front more dominant. This leads to the larger distance between the ice edge and the fronts when the fronts are more northward. In other words, the oceanic and atmospheric heat transport from the PF latitude to the south must compensate for the heat losses at a particular latitude in order to prevent sea surface temperature to reach the freezing point and sea ice formation. If the front is located at a more southward position, the surface temperature is colder and the oceanic heat losses can be very large. The heat transported southward of the PF are thus sufficient to prevent ice formation over a few degrees of latitude south of the PF only. This leads to a smaller distance between the PF and the ice edge than where the PF is more northward and thus in regions where the climate is milder. This effect can be modulated by the wind direction (this part will be modified in the revised version to make the links stronger, see the response to point 4 above). We also discuss this contribution of the atmospheric and oceanic heat transport from the front in the Results section (labelled subsection 3d in the revised section). However, we were not able to find a clear link between the latitude of the front and SST gradients or with EKE, the latter having too large longitudinal variations to identify a systematic effect of latitude.

Additionally, we propose to add the following text in the paragraph in which we explain the origin of the positive correlation to make this clearer:

Specifically, the heat transport from the latitude of the PF to the ice edge must compensate at all longitudes for the cooling (i.e. heat loss) at the ice edge, to prevent sea ice freezing. When the ice edge is located further to the south, where atmospheric temperatures tend to be colder, the oceanic heat loss is typically larger, and the heat transported from the PF should also be larger to prevent ice formation. This can be attained only over relatively short distances, such that the ice edge remains closer to the PF. In contrast, where the ice edge is located further north in a milder climate (with a warmer atmosphere), the heat required to prevent sea ice freezing is considerably lower. The PF can then be more distant from the ice edge, with the ocean still providing sufficient heat to avoid ice formation.

8. The Results section is quite extensive. Please consider dividing it into thematic subsections for readability.

   As suggested, we will add in the revised version subsections in the Results section.

9. Fig 2: Adding the winter sea ice edge would be useful for interpretation. Please also clarify which definition is adopted for the frontal positions (perhaps Orsi or Park) in the caption.

   The position of the winter ice edge will be added in the revised version. The front follows the definition of Orsi. This will be specified in the revised version.

10. Fig 4: Please unify the y-axis ticks for the wind velocity.

As suggested, we will unify the y-axis ticks for the wind velocity in the revised version.

11. Is Fig 8 supposed to be referred in somewhere around L409-420?

Sorry to have missed this reference. We will add in the revised version a reference to Fig. 8 when presenting the regression model.

12. Fig 9: Might be more effective and easier to interpret if presented in a normal plan view (top-down map perspective) rather than the current tilted view.

We agree that a normal plan view can be more effective but the advantage of the tilted view is to show clearly that this figure is a sketch, in contrast to the other figures in normal plan view. Nevertheless, we will reevaluate this point for the revised version and check which option is the most adequate.

I believe that addressing these points will strengthen the manuscript and offer a more balanced perspective on the complex oceanographic controls influencing the Antarctic winter sea ice edge. This work is otherwise of high quality and is well-suited for publication in The Cryosphere. I sincerely thank the authors for their valuable contribution and look forward to their response.

Thanks again for the constructive suggestions that will improve the quality of our manuscript and strengthen our conclusions.

---

## Referee Report (RR1)

Manuscript "Winter sea ice edge shaped by Antarctic Circumpolar Current pathways." Authors: Hugues Goosse, Stephy Libera, Alberto C. Naveira Garabato, Benjamin Richaud,

Alessandro Silvano, MarCn Vancoppenolle

I would like to thank the authors for taking into consideration my suggestion to extend the analysis regarding the selection of the meridional wind location. I think this addition provides a beneficial detail to the overall analysis.

The manuscript is very well written, and the corrections and modifications made have improved its clarity and comprehensiveness.

I would support its pubblication with just minor additional comments reported here.

**Minor comments**

Line 142-143. Since the definition of the SAF is given, and its contour is included in Fig.1, it should be mentioned why it is not included in the analysis

Line 176-193. The author say that the datasets considered cover different periods, but it would be good to specify if the total period of availability for each of them (for observations/reanalysis) has been used in the study or, if not, which period. It would give a more comprehensive idea of the period of overlapping among the datasets.

Line 294. "The main element that explains these changes in the distance.." I think would sound better

Line 299. The correlation for the SBdy front following the definition of Orsi et al., (1995) is almost always giving offset values with respect to the other two fronts and even to the same front following the definition of Park et al., (2019). Is there a possible explanation for that?

Figure 8/Table 3. It would be interesting to include in this analysis the other two fronts to see if the results are similar in terms of sea ice edge prediction.

Best Regards, Martina Zapponini

---

## Author Response (AR2)

**Response to the comments of Martina Zapponini**

I would like to thank the authors for taking into consideration my suggestion to extend the analysis regarding the selection of the meridional wind location. I think this addition provides a beneficial detail to the overall analysis. The manuscript is very well written, and the corrections and modifications made have improved its clarity and comprehensiveness.

I would support its publication with just minor additional comments reported here.

We would like to thank again the reviewer for the positive evaluation of this revised version and the suggestions, both at this stage and for the initial submission, that have contributed to improve the quality of our manuscript. Our responses to the comments are below in blue.

**Minor comments**

Line 142-143. Since the definition of the SAF is given, and its contour is included in Fig.1, it should be mentioned why it is not included in the analysis.

We mentioned lines 92-93 of the revised version that 'The SAF is located even farther north of the sea ice edge, and is therefore not expected to directly impact sea ice'. To make this point clearer, the text has been modified to 'The SAF is located even farther north of the sea ice edge. It is therefore not expected to directly impact sea ice and is not addressed here.'

Line 176-193. The author say that the datasets considered cover different periods, but it would be good to specify if the total period of availability for each of them (for observations/reanalysis) has been used in the study or, if not, which period. It would give a more comprehensive idea of the period of overlapping among the datasets.

For each of the dataset, we use the full period of availability at the time of our analysis. Several records have a small overlap or even none. For instance, the front definition of Orsi is based on in situ observations available up to 1990 while the definition of Park et al. uses satellite observations between 1993 and 2012. However, the mean signal on the mean position of the winter ice edge is robust and stable enough through time to identify links between the different variables even if different periods are selected. To make this more explicit, we insist more on the exact period covered by the observations in the revised version.

Line 294. "The main element that explains these changes in the distance." I think would sound better

Thanks. This has been modified in the new version.

Line 299. The correlation for the SBdy front following the definition of Orsi et al. (1995) is almost always giving offset values with respect to the other two fronts and even to the same front following the definition of Park et al., (2019). Is there a possible explanation for that?

The SBdy is the front that displays the largest differences between the two definitions (Orsi vs. Park), likely because its signature is weaker than the one of the other fronts (see the second paragraph of section 3a for instance). In the definition of Park, the SBdy position is more parallel to the one of the other fronts, probably because of the selected definition based on SSH. This could be the reason of the higher correlation but this is just a hypothesis and a more detailed comparison of the two definitions and an analysis of the limitation of the observations at those high latitudes would be required to verify this.

Figure 8/Table 3. It would be interesting to include in this analysis the other two fronts to see if the results are similar in terms of sea ice edge prediction.

The position of all the fronts is highly correlated with the position of the ice edge (Table 2) so by construction similar results can be obtained using any of them. The numbers could slightly change but this would not modify our conclusions. To avoid a too long discussion, we prefer to include the analysis for only one of the fronts (i.e. the PF).